# Large-scale simulation of biomembranes incorporating realistic kinetics into coarse-grained models

Mohsen Sadeghi [1✉] & Frank Noé [1✉]

Biomembranes are two-dimensional assemblies of phospholipids that are only a few nanometres thick, but form micrometre-sized structures vital to cellular function. Explicit molecular modelling of biologically relevant membrane systems is computationally expensive due to the large number of solvent particles and slow membrane kinetics. Coarse-grained solvent-free membrane models offer efficient sampling but sacrifice realistic kinetics, thereby limiting the ability to predict pathways and mechanisms of membrane processes. Here, we present a framework for integrating coarse-grained membrane models with continuum-based hydrodynamics. This framework facilitates efficient simulation of large biomembrane systems with large timesteps, while achieving realistic equilibrium and non-equilibrium kinetics. It helps to bridge between the nanometer/nanosecond spatiotemporal resolutions of coarse-grained models and biologically relevant time- and lengthscales. As a demonstration, we investigate fluctuations of red blood cells, with varying cytoplasmic viscosities, in 150-milliseconds-long trajectories, and compare kinetic properties against single-cell experimental observations.

[1] Department of Mathematics and Computer Science, Freie Universität Berlin, Arnimallee 6, 14195 Berlin, Germany. ✉email: mohsen.sadeghi@fu-berlin.de; frank.noe@fu-berlin.de

ipid bilayers are essential structural elements of living cells, and are important for various cellular functions, such as exo/endocytosis and signal transduction[1,2]. The unusual properties of biomembranes—they are two-dimensional fluids, but have an out-of-plane elasticity similar to solid sheets—have been the subject of numerous biophysical studies during the past decades[3,4]. Nonetheless, simulating biologically relevant membrane systems remains a challenge[5]. Considering length- and timescales involved in biological processes, coarse-graining has become an essential approach in membrane simulations[6,7]. Interacting particle reaction-dynamics (iPRD) models[8] are extremely coarse-grained models that have been used to simulate a wide variety of cellular signalling pathways[9,10]. In such simulations, systems containing proteins, lipids, and metabolites are modelled with particles large enough to represent whole proteins or a patch of membrane lipids[11].

While many coarse-grained membrane models may reproduce correct equilibrium properties[12,13], membrane kinetics has been treated much less satisfactorily. Correct kinetics are essential in order to make predictions about not only how fast, but also by which mechanisms biological processes happen: considering that biological functions are far from thermodynamic equilibrium[14,15], the pathways by which systems relax to steady states or transition between them directly depends on the kinetics. As examples, consider passive transport of molecules through the membrane, driven by density fluctuations[16], dynamics of membrane scission by ESCRT proteins[17], as well as dynamin super-family[18], and questions such as whether vesicle exo/endocytosis rather proceeds by fusion and recycling or by a faster partial fusion (kiss and run) mechanism[19,20]. Membrane kinetics is thus indispensable when investigating membrane-mediated interactions, association/dissociation events and membrane remodelling processes. Also, considering that novel experimental techniques such as dynamic optical displacement spectroscopy (DODS) have made it possible to look at membrane fluctuations resolved at 20 nm and 10 μs range[21], modelling tools that are up to the task of combining the large-scale dynamics of the membranes, while resolving these microscopic scales are needed more than ever.

Membrane-solvent coupling is essential to correct membrane kinetics, rendering the issue more pronounced in the so-called solvent-free membrane models, in which the interactions of membrane particles are adjusted, so as to implicitly account for the missing solvent[22,23]. Discarding solvent particles drastically reduces the computational cost, but leads to unrealistically fast kinetics. A simple correction is the so-called time-mapping[24], i.e. to artificially scale the time in order to match the experimental value of a specific kinetic property, such as lipid diffusion[25]. But this approach fails when multiple timescales are present and can therefore not improve our ability to predict mechanisms, as it preserves the relative order between timescales. An alternative is to use a simplified coarse-grained explicit solvent[26,27], or to use the lattice Boltzmann method to couple particle motions to a grid-based numerical solution of fluid dynamics[28]. Both approaches limit the accessible time- and lengthscales, due to additional computations necessary for the fluid response.

Considering both the need for reliable kinetics, and the wide range of scales, a desirable solution for a solvent-free membrane model would be to implicitly incorporate the two major timescales corresponding to in-plane and out-of-plane motions. For this purpose, we propose to use anisotropic stochastic dynamics with hydrodynamic interactions. Yet, instead of relying on ad hoc descriptions of friction and diffusion, or a posteriori time-mapping, we derive the governing dynamics of membranes coupled with fluid environments from first principles. We demonstrate how this approach results in the expected equilibrium and non-equilibrium kinetics. Furthermore, we demonstrate its efficiency

and robustness in modelling large-scale dynamics of a realistic biological setup by simulating a human red blood cell and obtain ~150 ms single-trajectories. We show how observables such as power spectral density of cell thickness fluctuations match the experimental measurements, offering potential applications in establishing a link between simulation and experiment in single-cell profiling and diagnosis.

## Results

**Anisotropic stochastic dynamics.** Figure 1a shows a generic coarse-grained model of the bilayer membrane in which the two leaflets are resolved. In the stable bilayer phase, the particles comprising the membrane diffuse laterally in the membrane domain, while they are coupled to the hydrodynamics of the solvent domain in the out-of-plane direction. The discretized equations of motion of these particles can be written down in a very general form, using anisotropic over-damped Langevin dynamics with hydrodynamic interactions[29],

$$\Delta \mathbf{r}_i = \Delta t \sum_j \nabla_j \cdot \mathbf{D}_{ij} + \frac{\Delta t}{kT} \sum_j \mathbf{D}_{ij} \cdot \mathbf{F}_j + \boldsymbol{\chi}_i(\Delta t) \qquad (1)$$

where subscripts $i$, $j$ are particle indices, $\mathbf{D}_{ij}$ is the diffusion tensor, $\mathbf{F}_j$ is the sum of forces acting on the $j$-th particle, $\nabla_j \cdot$ denotes the divergence with respect to the coordinates of the $j$-th particle, $k$ is the Boltzmann constant and $T$ is the temperature. The noise term, $\boldsymbol{\chi}_i(\Delta t)$, is the outcome of a Gaussian process described, in one timestep, by the moments,

$$\langle \boldsymbol{\chi}_i(\Delta t) \rangle = 0 \qquad (2)$$

$$\langle \boldsymbol{\chi}_i(\Delta t) \boldsymbol{\chi}_j(\Delta t) \rangle = 2 \, \mathbf{D}_{ij} \Delta t \qquad (3)$$

and uncorrelated between subsequent steps. Several approximations of the $\mathbf{D}_{ij}$ tensor exist, mainly for spherical particles floating freely in solvents. Starting from the Stokes–Einstein model, $\mathbf{D}_{ij} = \frac{kT}{6\pi\eta R} \delta_{ij} \mathbf{I}$[30,31] (with $\eta$ being the viscosity of the solvent and $R$ the particle radius) to the more sophisticated models such as the Oseen[32] and Rotne–Prager–Yamakawa[33] tensors. These models are constructed based on analytic solutions to continuum hydrodynamics, albeit with simplifying assumptions, and can be used to include hydrodynamic interactions between particles. But if we consider such a description for densely-packed particles forming a membrane, which usually partitions the space into interior and exterior regions, and through which the solvent cannot easily permeate, it is obvious that a new solution is required.

We construct an anisotropic hydrodynamic description via considering a local orthonormal basis at the outer surface of one of the membrane leaflets, and decomposing the displacement of each particle as the sum of in-plane and out-of-plane contributions (Fig. 1a). Thus, the in-plane dynamics is dictated by the viscosity of the bilayer membrane, whereas the out-of-plane dynamics involves forces generated due to membrane elasticity and the dissipation through the fluid domain in which the membrane is suspended. The viscosity of these two environments differ by 2–3 orders of magnitude[34,35]. As a result, simulation schemes operating on one timescale cannot reproduce the kinetics efficiently. The main ideas for building an efficient and accurate model for the anisotropic dynamics of membrane particles are: (i) The main contribution to solvent-mediated hydrodynamic forces acts along the membrane normal. While shearing interactions via the solvent can affect in-plane diffusion[36,37], they are generally dominated by much larger in-plane viscous forces, especially in highly coarse-grained models. Also, they can be neglected, when large-scale out-of-plane dynamics are considered. Finally, if there are other mechanisms

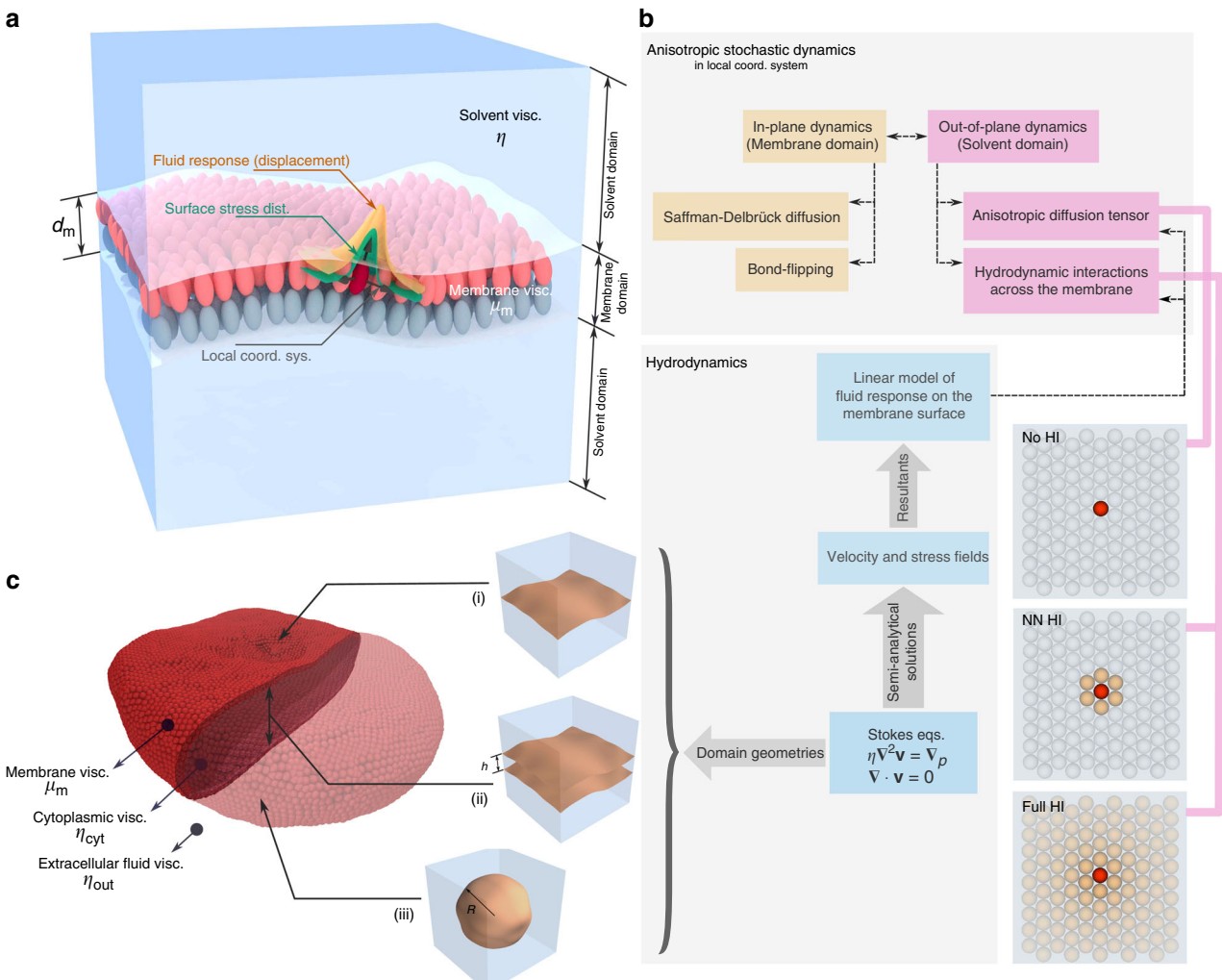

**Fig. 1 Overview of the proposed method for anisotropic stochastic dynamics of a coarse-grained membrane model. a** Schematic of a coarse-grained membrane model with the two leaflets resolved. For a membrane suspended in solvent, distinct membrane and solvent domains are designated. The local coordinate system describing the in-plane and out-of-plane directions, as well as the distributed boundary conditions and the fluid response are shown for a selected particle. **b** Schematic of the proposed method for handling anisotropic stochastic dynamics. **c** Three distinct idealized geometries used in the derivation of the fluid response: (i) single planar membrane, (ii) parallel planar membranes, (iii) spherical vesicle. The cell membrane of a human red blood cell can be considered to experience fluid responses approximated by a combination of the three geometries.

controlling the in-plane diffusion, such as the bond-flipping Monte Carlo moves[11,38], the contribution from shearing interactions becomes redundant. (ii) While in-plane hydrodynamics of bilayer membranes can also be studied rigorously[39], a highly coarse-grained membrane model would benefit little from it. Also, for in-plane diffusion in a membrane crowded with proteins, there is evidence pointing to the hydrodynamics being effectively reduced to a collision-based dynamics, resulting in a Stokes–Einstein-like diffusion[40]. Based on these arguments, we propose the following form for the diffusion tensor,

$$\mathbf{D}_{ii} = D_i^{\parallel}\mathbf{I} + (D_{ii}^{\perp} - D_i^{\parallel})\mathbf{n}_i\mathbf{n}_i$$

$$\mathbf{D}_{ij} = D_{ij}^{\perp}\mathbf{n}_i\mathbf{n}_j, \quad i \neq j \qquad (4)$$

where $D^{\parallel}$ and $D^{\perp}$, respectively, represent the in-plane and the out-of-plane diffusion coefficients and $\mathbf{n}_i$ is the unit vector normal to the membrane surface at the position of the $i$-th particle. Provided that the values of $D^{\parallel}$ and $D^{\perp}$ are known for all the particles forming the membrane, we can make use of Eq. (1) to efficiently obtain particle trajectories.

As the breakdown chart in Fig. 1b suggests, we use the Saffman–Delbrück model of the diffusion of cylindrical inclusions in fluid sheets to obtain the in-plane components, $D_i^{\parallel}$ (Eq. (11))[41,42]. Other choices are also possible, provided that they are consistent with the macroscopic viscosity of the membrane (see "In-plane diffusion coefficient" under Methods section, and also ref. [43] for a discussion on how to make this connection). However, there are no readily available descriptions of hydrodynamics yielding out-of-plane components, $D_{ii}^{\perp}$ and $D_{ij}^{\perp}$. To derive numerical values for these components, we have developed semi-analytical solutions to the Stokes equations in select idealized geometries (Fig. 1b and c)[43].

The underlying approach is to use Gaussian distributed velocity or stress boundary conditions as the test input, and numerically integrate the resulting fields over selected patches on the surface of the membrane, i.e. find the generated forces or displacements which describe the fluid response (Fig. 1a). For the simple case of a single planar membrane, we have found a closed form solution for $D_{ii}^{\perp}$[43],

$$D_{ii}^{\perp} = \frac{kT}{8\sqrt{\pi}\eta\alpha} \times e^{-\xi_c}[I_0(\xi_c) + I_1(\xi_c)] \qquad (5)$$

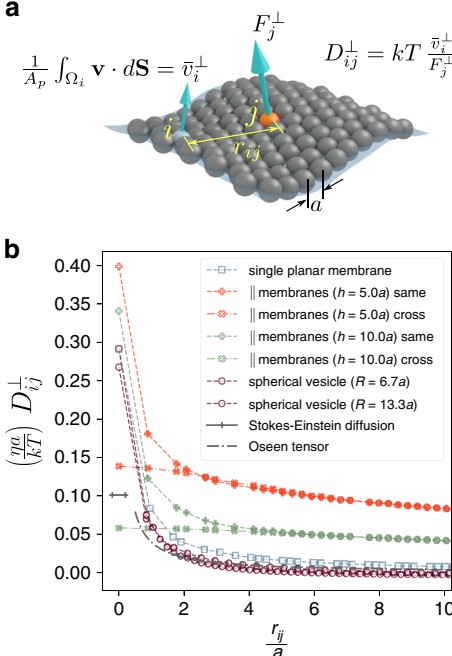

**Fig. 2 Numerical calculation of the out-of-plane component of the diffusion tensor, $D_{ij}^{\perp}$. a** Definition of the $D_{ij}^{\perp}$ based on $F_j^{\perp}$, the force existing between the fluid domain and particle $j$, and the corresponding effective out-of-plane velocity, $\bar{v}_i^{\perp}$, attributed to particle $i$. The effective velocity results from averaging the velocity field of the fluid, in the direction normal to the membrane, in the vicinity of the membrane particles. **b** Compilation of numerical values of the $D_{ij}^{\perp}$ as a function of in-plane inter-particle distance, $r_{ij}$. Results are, respectively, given for a single planar membrane, two sets of parallel membranes with the given inter-plane separations, and two spherical vesicles with the given radii. In the case of parallel membranes, the in-plane distance is measured solely based on planar coordinates, and for spherical vesicles, it is defined along the geodesics. For these calculations, the parameter $\alpha$ is chosen equal to $0.1a$. Values given by the Stokes–Einstein relation, as well as the Oseen tensor are shown for comparison.

where $\xi_c = \frac{A_p}{8\pi\alpha^2}$, with $\alpha$ being a scaling parameter corresponding to the width of the Gaussians used as input, $A_p$ is the area per particle, and $I_0$ and $I_1$ are the modified Bessel functions of the first kind. The problem is not, of course, tractable for arbitrary geometries. However, based on the solutions presented in ref. [43], it is possible to find numerical results specific to a membrane model, using the scheme depicted in Fig. 2a. We have derived the results for several prototypical membrane geometries, based on our previously developed membrane model (Fig. 2b)[11,43].

Comparing values of $D_{ij}^{\perp}$ with the diffusion coefficient given by the Stokes–Einstein formula, and $D_{ij}^{\perp}(r_{ij})$ with the magnitude of hydrodynamic interactions predicted by the Oseen tensor (Fig. 2b), demonstrates the shortcoming of both descriptions when applied to membrane particles. Interestingly, compared with free-floating spherical particles described by the Stokes–Einstein model, membrane particles have a much higher mobility in the out-of-plane direction. This could be explained by noting that the solvent only affects the particles in the membrane normal direction. Also, the Oseen tensor can only be used to approximate the asymptotic behaviour of hydrodynamic interactions over long ranges up to a multiplicative constant.

Comparing the results for different geometries given in Fig. 2b, two conclusions are in order: (i) Membrane curvature has little effect on hydrodynamic interactions. This can be readily observed

in Fig. 2 by comparing the results corresponding to single planar membrane, and the two spherical vesicles with different radii. The deviation is only important for $D_{ii}^{\perp}$ components. (ii) Particles on parallel membranes have generally larger $D_{ii}^{\perp}$ and $D_{ij}^{\perp}$ values compared with single membranes, and there also exists non-negligible hydrodynamic interaction between particles on the two facing leaflets. This higher mobility can be considered as the cause for fluctuation-magnification observed for membranes near walls[44]. The closer the membranes are together, the more pronounced are the confinement effects.

Finally, one important aspect of the proposed method, which drastically distinguishes it from previous descriptions such as the Oseen tensor, is that the hydrodynamic interactions between particles are calculated in the presence of the rest of the particle system in a given geometry. While in more generic methods, due to the complexities arising from recurrent interactions between particle pairs, triplets, etc. enhancing the approximation to include such effects is immensely difficult.

**Kinetics of a planar membrane at equilibrium**. Based on the introduced diffusion tensor, we investigate the kinetics of equilibrium membrane fluctuations for planar membranes coupled to an aqueous solvent with the viscosity of $\eta = 0.890$ mPa s. Here, we employ the membrane model introduced in ref. [11], where the bilayer consists of closely-packed laterally mobile particle dimers (see "Mesoscopic membrane model" under Methods section for details), but other coarse-grained membrane models[6,7] could be used as well. It is well-known that hydrodynamic interactions are generally long-ranged[45], but it is worth noting that in contrast to systems of free particles, the hydrodynamic interactions between membrane particles have to compete with large forces resulting from the bending rigidity of the membrane. To investigate how well local and longer-range estimates of hydrodynamic effects compare, we consider three different models of hydrodynamic interactions (Fig. 1b): (i) No HI: No hydrodynamic interactions exist between membrane particles. The only non-zero diffusion coefficients are $D_i^{\parallel}$ and $D_{ii}^{\perp}$, whose values are correspondingly taken from Eqs. (11) and (5)) This results in a block diagonal matrix of assembled diffusion tensors and local anisotropic dynamics, with no pairwise correlation between particles as described by Eq. (3). (ii) NN HI: Hydrodynamic interactions are only included between nearest-neighbour particles that are defined by construction of our membrane model, resulting in a fast implementation. To introduce correlated random displacements resulting from these interactions, we construct local diffusion matrices and use Cholesky decomposition to transform a vector of independent normally distributed random variables to a correlated vector obeying Eq. (3). (iii) Full HI: Hydrodynamic interactions are implemented across the membrane, with global diffusion matrices compiled in each iteration. To avoid the $\mathcal{O}(N^3)$ cost of a full Cholesky decomposition, we use the approach developed by Geyer and Winter to approximate the correlated random forces[46].

Further details of the simulations are given in "Simulation of planar membranes" under Methods section. Based on the chosen force field, and calculated diffusion tensors, we found a timestep of 0.5 ns to produce stable trajectories. Compared with timesteps of at most 20 ps for the same model without hydrodynamics[11], there is an apparent 25-fold increase in performance. Though the cost of integrating stochastic dynamics in each model of hydrodynamics is to be considered. With the cheap No HI model, the speed-up significantly enhances the timescales available to exploration with the current model, pushing simulation times to the 100 ms range.

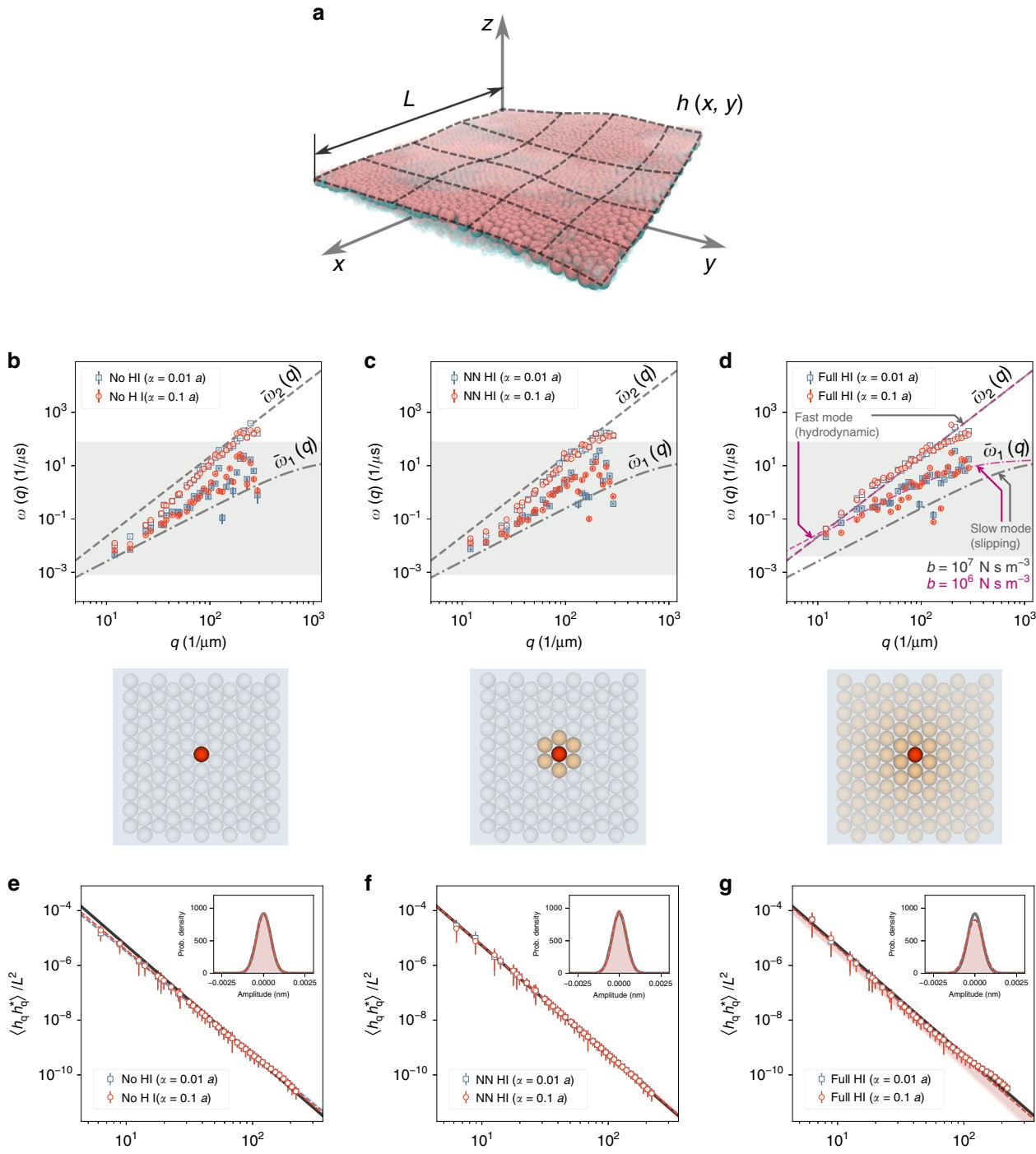

**Fig. 3 Simulating equilibrium thermal undulations of a planar membrane suspended in water. a** An overlay of several simulation snapshots performed using the mesoscopic membrane model with the lattice parameter of $a = 10$ nm. Also shown is a schematic depiction of the smooth height function, $h(x, y)$ fitted to particle positions in each frame. **b–d** Dispersion relations for the membrane, calculated based on the assumption that the fluctuations in the amplitude corresponding to a wave vector **q** relax biexponentially with frequencies $\omega_{1,2}(q)$. The fast regime (higher $\omega$) is depicted with empty symbols, while filled symbols are used for the slow regime. Results are shown for two different choices of the scaling factor $\alpha/a$. Predictions of the continuum model of Seifert et al. ($\bar{\omega}_{1,2}$ functions from Eqs. (15) and (16)) are included for comparison. In figure **d**, the magenta curves correspond to the same continuum model with a different inter-leaflet friction coefficient (see "Equilibrium undulations and dispersion relations" under Methods section). The light grey shaded region signifies the range of frequencies available, depending on the sampling rate and the length of the trajectories. The three cases correspond, respectively, to the No HI, NN HI and Full HI hydrodynamic models. **e–g** Power spectra of thermal undulations of the membrane patches in the aforementioned simulations. Dashed lines are fits of the function $C(qL)^n$ to the data, whereas the solid black line is the prediction of the continuum model given by Eq. (13). Shaded areas denote 99% prediction intervals of the power-law fits. The inset panes are plots of the probability distribution of the real and imaginary parts of $h_\mathbf{q}$, binned together, for the short-wavelength mode $\mathbf{q} = \frac{2\pi}{L}(50, 50)$. This mode mostly corresponds to the out-of-plane motion of individual particles and is thus the fastest dynamical mode of the system. The grey curve is the Gaussian distribution predicted by the Helfrich model (see "Equilibrium undulations and dispersion relations" under Methods section). Error bars in all plots represent standard deviations.

In all cases, the power spectra of thermal undulations follow the expected equilibrium behaviour of Eq. (13) (Fig. 3e–g). This result indicates that (a) we have indeed sampled from equilibrium configurations of the membrane, and (b) fluctuation–dissipation theorem holds for the stochastic dynamics developed here. The latter hinges upon the validity of the general form of the diffusion tensor (Eq. (4)) as well as correct approximation of noise terms (Eq. (3)). Also, the probability distribution of the fastest dynamical mode of the system (inset plots of Fig. 3e–g) is a Gaussian with the variance of $kT/\kappa L^2 q^4$ (see Eq. (14)), indicating that the large timestep chosen here is suitable for equilibrium sampling.

The equilibrium kinetics of membrane fluctuations can be investigated by measuring the relaxation time of the thermally-induced undulations. This mode-dependent relaxation dynamics yields the so-called dispersion relation of the membrane. A reliable theoretical description, that has also been shown to be consistent with experiments, is given by the continuum model of Seifert et al.[47,48]. This model predicts the relaxation dynamics of each undulatory mode, with the wave vector $\mathbf{q}$, to follow a biexponential decay (see "Equilibrium undulations and dispersion relations" under Methods section for details). The theoretical values for the two corresponding frequencies are denoted here as $\bar{\omega}_{1,2}(q)$ (dashed lines in Fig. 3b–d). The asymptotic values of these two frequencies are, in our range of inspection, given as $\bar{\omega}_1(q) \approx \frac{K_{area}}{4b}\frac{\kappa}{\kappa}q^2$ (slow mode) and $\bar{\omega}_2(q) \approx \frac{\tilde{\kappa}}{4\eta}q^3$ (fast mode)[49]. The slow, or slipping, mode corresponds to in-plane density fluctuations, as well as the friction between membrane leaflets, while the fast, or hydrodynamic mode is due to the viscous loss in the fluid. The slow mode depends strongly on the internal dynamics of the membrane, and thus, on the details of the coarse-grained model, while the fast mode is only a function of membrane rigidity and the hydrodynamics of the solvent, making it our focus of interest here.

Our simulation results, especially with the Full HI model, compare remarkably well with the expected dispersion relations (Fig. 3b–d) and exhibit very good approximations of the desired fast (hydrodynamic) mode. This proves that the framework laid out here is indeed capable of yielding trajectories with the realistic kinetics of the membrane-solvent system, over a wide range of spatial and temporal frequencies. Interestingly, the slow (slipping) mode is also present, and within the range expected based on the parameters fed to the model (see "Equilibrium undulations and dispersion relations" under Methods section). The choice of the hydrodynamic scale parameter, $\alpha$, at least in the range of inspection, has little effect on the simulated dispersion relations. This will prove helpful in robust application of this framework to other membrane models and across different scales. While the Full HI model best reproduces the fast mode, even without the inclusion of hydrodynamic interactions, the viscous dissipation is captured rather well in the No HI model. This allows for efficient equilibrium sampling of membrane dynamics with coarse-grained models, relying solely on a locally anisotropic stochastic dynamics.

**Non-equilibrium relaxation dynamics**. The next step is to investigate the dynamics of the membrane-solvent system, starting from states far from thermodynamic equilibrium. We consider how an initially flat membrane, coupled to solvent hydrodynamics, evolves towards equilibrium. Arguably, there exists no theoretical description similar to Eq. (15) to describe this non-equilibrium evolution. Thus, we use a generic exponential relation of the form of $\frac{kT}{\kappa (qL)^4}(1 - \exp(-t/\tau))$ to describe

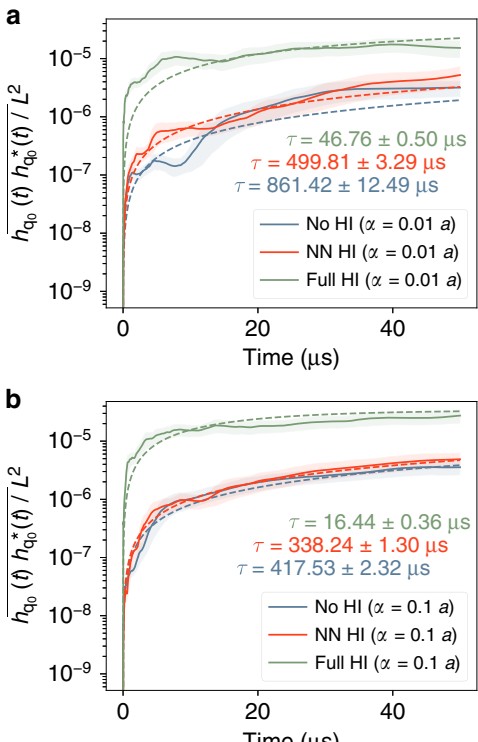

**Fig. 4 Relaxation of the energy of the largest undulation mode, with the wave vector $\mathbf{q_0} = \frac{2\pi}{L}(\mathbf{1, 0})$, for an initially flat membrane, evolving towards equilibrium.** Results are given for all the hydrodynamic cases, using two scaling factors **a** $\alpha = 0.01a$ and **b** $\alpha = 0.1a$. Dashed lines are fits of the function $\frac{kT}{\kappa (qL)^4}(1 - \exp(-t/\tau))$ to the data. Corresponding values of the timescale $\tau$ for each case are colour-coded. Shaded error bands represent one standard deviation.

the time evolution of the energy of an undulatory mode, and use the parameter $\tau$ as a relaxation time (Fig. 4a and b).

While equilibrium membrane kinetics are mostly insensitive to the range to which the hydrodynamic interactions are present (Fig. 3), this clearly affects the non-equilibrium kinetics (Fig. 4). The No HI model has the slowest, and the Full HI the fastest dynamics towards equilibrium (Fig. 4a and b). If we compare the values of $\tau$ with the theoretical timescale of fluctuations of the same mode in equilibrium, which is 20.11 μs (based on the model described in Methods), the difference between the equilibrium and non-equilibrium dynamics becomes apparent. Only for the Full HI model the relaxation times are on the same order of magnitude.

**Non-equilibrium steady-state kinetics of active membranes**. As a more pronounced exampled of non-equilibrium dynamics, we investigate non-thermal undulations of the so-called active membranes[50], in which membrane-bound or transmembrane proteins exert forces on the membrane in the normal direction, consuming chemical or light energy to induce local displacements. Prominent examples are the light-driven bacteriorhodopsin (bR) proton-pump proteins[51]. Theoretical models based on stochastic active forces in this system and the resulting non-equilibrium undulations have been proposed[52–54]. Here we choose a description, closest to the formulation presented in ref. [54], with the reaction,

$$\text{UP} \underset{k_{on}}{\overset{k_{off}}{\rightleftharpoons}} \text{OFF} \underset{k_{off}}{\overset{k_{on}}{\rightleftharpoons}} \text{DOWN} \qquad (6)$$

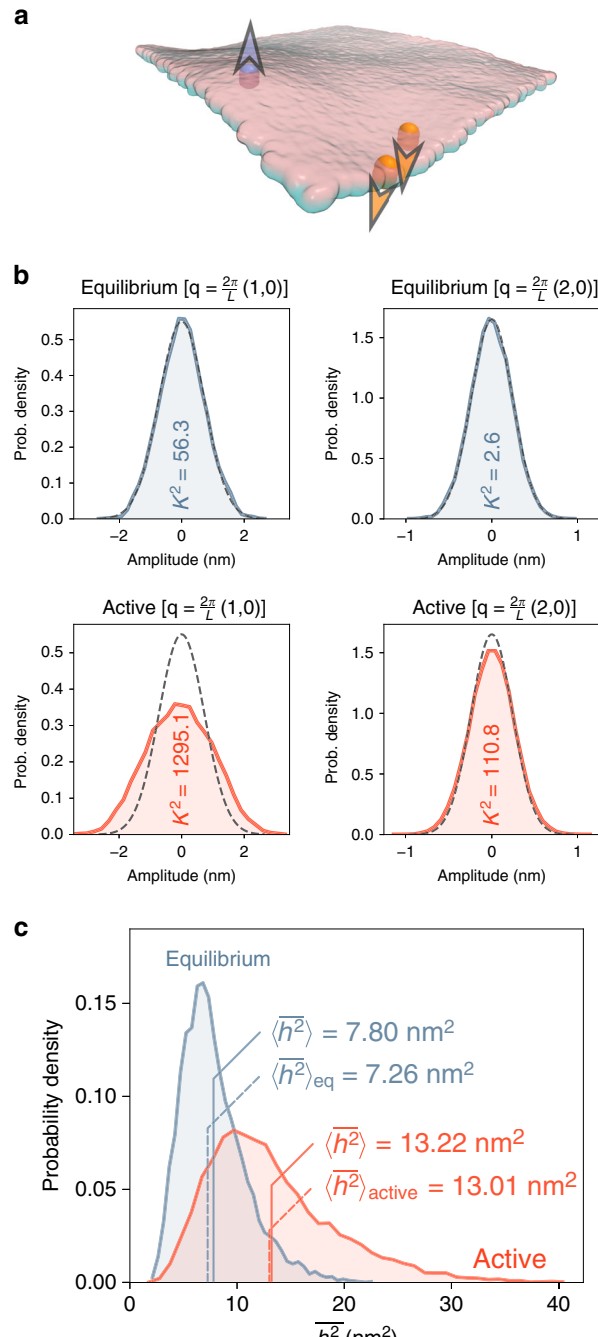

**Fig. 5 Non-equilibrium steady-state kinetics of membranes containing active pump proteins. a** Snapshot of the membrane and three active sites in the UP and DOWN states. **b** Distributions of the amplitude of two undulatory modes of the membrane with the given wave vectors **q**, in thermal equilibrium (blue) as well as non-equilibrium steady state with active proteins (red). Both real and imaginary parts of $h_{\mathbf{q}}$ are binned together. Dashed lines are Gaussians fitted to the equilibrium distribution, repeated in the second-row plots for comparison. The $K^2$ values from D'Agostino–Pearson test for the normality of the distribution[55], which are based on sample kurtosis and skewness, are given for each case. **c** Distribution of the squared height of the membrane, averaged across the projected area, for the equilibrium and active cases. Values of the mean-squared out-of-plane displacement of the membrane, $\langle \overline{h^2} \rangle$, for both equilibrium and active cases, are compared with the corresponding theoretical predictions $\langle \overline{h^2} \rangle_{eq}$ and $\langle \overline{h^2} \rangle_{active}$ (Eqs. (19) and (20)).

describing the stochastic three-state behaviour of active forces. Further details of the model are given in Methods.

We first use the model without active forces to obtain equilibrium undulations. Not surprisingly, the simulation results yield the expected Gaussian probability distribution for the amplitude of the two largest-wavelength undulatory modes (Fig. 5b and Eq. (14)). When active forces are present in the simulation, the height distribution deviates visibly for the large-wavelength modes (Fig. 5b). Non-Gaussianity of the height distribution is a hallmark of active processes[50] and is the result of active forces constantly driving the system out of equilibrium. For the distributions shown in Fig. 5b, comparing $K^2$ values of D'Agostino–Pearson normality test[55] quantitatively demonstrates non-Gaussianity when active forces are present.

To verify quantitatively that the model described here brings about correct non-equilibrium behaviour, we compare the mean-squared out-of-plane displacement, $\langle \overline{h^2} \rangle$, with the theoretically predicted values, $\langle \overline{h^2} \rangle_{eq}$ and $\langle \overline{h^2} \rangle_{active}$ (see "Active undulations" under Methods section for further details). The distribution of the squared displacement of the membrane is an observable which clearly differentiates equilibrium and non-equilibrium dynamics (Fig. 5c). The remarkable agreement of the mean-squared displacements values with the theoretical predictions (Fig. 5c), demonstrates how the presented model achieves accurate sampling in non-equilibrium steady-state scenarios.

**Fluctuation profile of a human red blood cell**. To demonstrate our membrane dynamics framework in a complex, biophysically relevant example, we consider a human red blood cell. Aside from being the classical subject of membrane biophysics studies[56–58], there has been a rather recent interest in profiling individual red blood cells for diagnostic purposes[59–64]. Diseases such as hereditary spherocytosis and sickle cell change the mechanics of the red blood cell membrane, as well as the rheology of its cytosol. They can thus be diagnosed by looking at the fluctuations of individual red blood cells using phase-shift microscopy techniques[60,65]. Another example is the pathology of the malaria disease, which is closely related to the mechanical response of the red blood cell to the invasion of parasite's merozoites[66–68]. Finally, the simple picture of the red blood cell thermal flicker has been revised by showing that there exist ATP-dependent active mechanisms, possibly reorganizing the spectrin cytoskeletal structure, leading to deviations from the expected equilibrium picture for timescales beyond 100 ms[21,69–71].

We have developed a particle-based model of the healthy human red blood cell (Fig. 6a) based on a 3D mesh obtained from refractive index tomography (further details of the model and the simulation setup are laid out in Methods). We have assigned different dynamics to the leaflets facing the inside and the outside of the cell, with the former as a planar surface in contact with a large body of aqueous solvent, and the latter as parallel surfaces with the inner space containing the more viscous cytoplasmic matrix. Our aim here is to show how different kinetics affects observables such as the magnitude of cell vibrations. Thus, while it is relatively straightforward to add active components to the model, similar to the results shown in Fig. 5, we have refrained from doing so to reduce the complexity of the model. Also, as mentioned, these active contributions are important when timescales beyond 100 ms are considered, while the longest trajectories presented here, though of significant length for such a nanometer/nanosecond whole-cell simulation, can still be considered within this passive regime.

The simulated distribution of the mean thickness over the cell's surface quantitatively matches the phase-shift imaging data (Fig. 6b), corresponding to a coherent representation of the cell

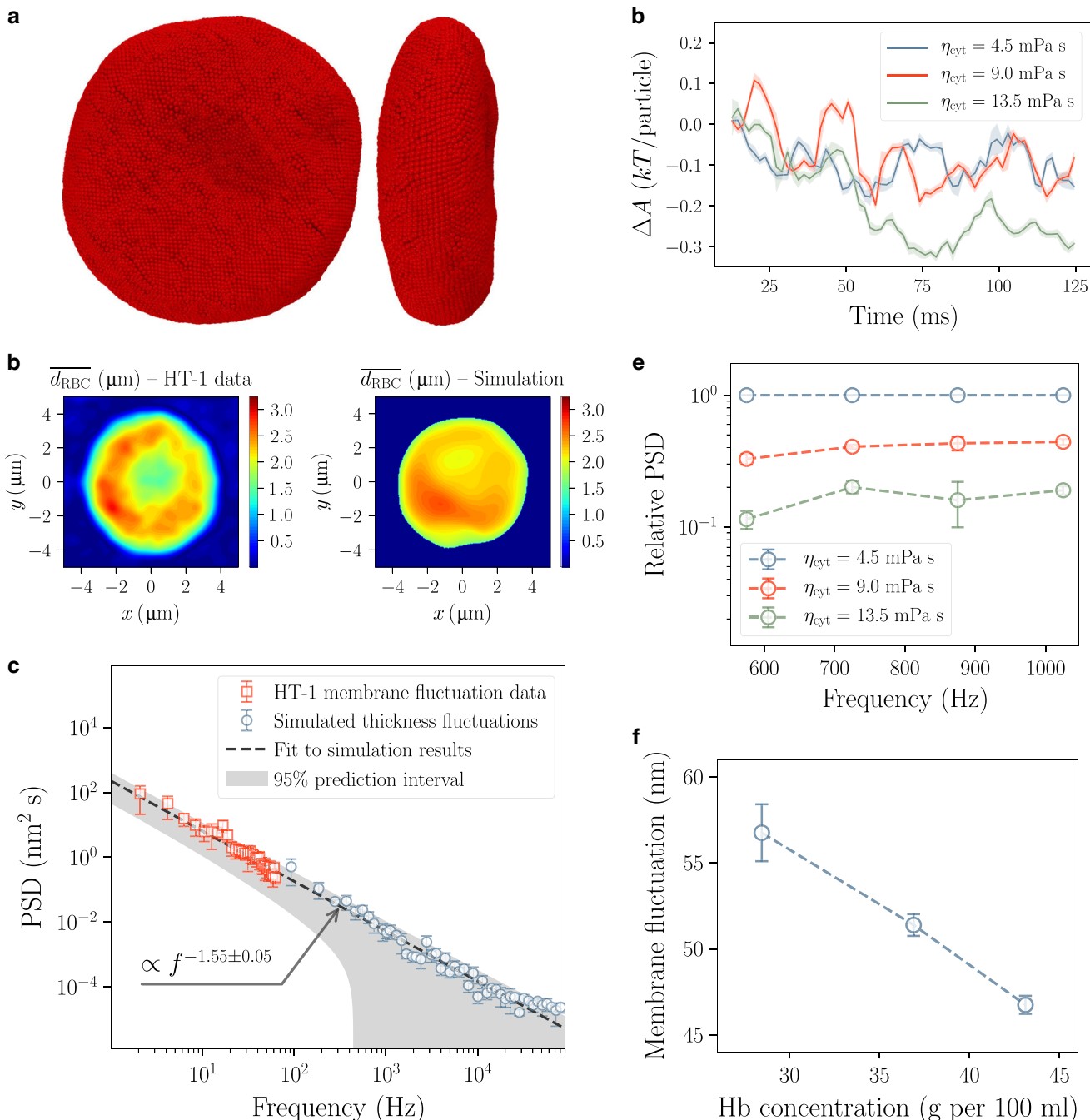

**Fig. 6 Simulating the dynamics of a human red blood cell. a** Top and side views of a snapshot of the red blood cell as modelled with the mesoscopic membrane model. **b** Mean cell thickness profiles from experimental measurements (holotomography and phase-shift imaging with HT-1 microscope, Tomocube Inc., Republic of Korea) compared with the simulated counterpart. **c** Power spectral density (PSD) of thickness fluctuations of the red blood cell for simulation and HT-1 experimental data. The power law is fitted to the simulation results and extended to the experimental low-frequency range for the sake of comparison. The shaded area corresponds to the prediction interval of the power-law fit. **d** Free energy change during the first 125 ms of simulations as a function of time, given for three different choices of the viscosity of the cytoplasm. **e** Relative PSD for the given viscosities, using the lowest viscosity of $\eta = 4.5$ mPa s as the baseline. **f** Magnitude of large-amplitude membrane fluctuations versus the haemoglobin concentration of the blood cell cytosol. Error bars or error bands represent standard deviations.

throughout the simulation. We compare the power spectral density (PSD) of thickness fluctuations with two observations. First, the power-law fit to the PSD versus frequency data from the simulation (dashed line in Fig. 6c), has the exponent of $-1.55 \pm 0.05$. A similar power law has been observed for healthy human red cells[56,70]. Theoretical models based on small-amplitude fluctuations yield an exponent of $-5/3$[72], while Brochard and Lennon's classic experiments put the exponent in the range $-1.45$ to $-1.3$ for $f > 1$ Hz[56]. Second, we have used single-cell phase-shift imaging data of five healthy red blood cells, and have calculated the corresponding PSD values (Fig. 6c). We have demonstrated the agreement between simulation and experiment by extending the power-law fit to cover the experiment's frequency range (Fig. 6c).

We have repeated the simulation for different values of cytoplasmic viscosity (Fig. 6d–f). The non-equilibrium evolution

of the Helmholtz free energy of the red blood cell exhibits the expected descent towards minimum in all cases (Fig. 6d), but it is obvious that the path taken in the free energy landscape is significantly different for the two lower-viscosity cases compared with the high-viscosity one. This result points to the fact that different kinetics can push the system along different configurational pathways, potentially altering path-dependent phenomena.

Finally, to demonstrate how the dynamical framework developed here can pertain to quantities important in diagnosis, we have compared the values of power spectral density corresponding to a small range of frequencies, when the cytoplasmic viscosity changes (Fig. 6e). The significant change in PSD with viscosity is easily measurable based on the fluctuation spectrum. This deviation could potentially point to an anomaly in a diseased cell. To illustrate this point, we have compared the amplitude of large-wavelength membrane fluctuations when the haemoglobin (Hb) concentration of the cytoplasm changes. The seemingly linear correlation between the membrane fluctuation and Hb concentration (Fig. 6f) has been previously suggested based on quantitative phase imaging techniques[60,65]. Specifically, Lee et al. have shown that in patients with diabetes mellitus, the change in the concentration of haemoglobin in red blood cells has a statistically significant effect on membrane fluctuations, with a correlation very similar to Fig. 6f [65].

## Discussion
With the introduction of a framework based on diffusion tensors that fully describe anisotropic dynamics as well as hydrodynamic effects, we have implemented a robust method to tackle the two-scale kinetics of a coarse-grained membrane model. The general form of the diffusion tensor introduced here (Eq. (4)) can be used beyond the idealized geometries presented here (Figs. 1c and 2b). Our approach can also be integrated into numerical grid-based methods, such as the lattice Boltzmann, using iterative linearisation of the fluid response, when the membrane acquires complex geometries. This would still introduce a significant computational gain, as the grid-based method only needs to be invoked when a significant change in geometry or environment is detected. Furthermore, the hydrodynamic interaction model presented here inherently possesses a length scale, allowing for membrane-bound species of different lateral size in one coherent framework.

The equilibrium kinetics results demonstrate the correct balance between fluctuating thermal energy of the heat bath and hydrodynamic dissipation. The consistency of non-equilibrium steady-state results extends the range of this balance to where an influx of energy from a non-thermal source is dissipated. We have thus confirmed both to be equally well represented by the same approach. This outcome is important for the simulation of membrane processes in living cells that follow similar non-equilibrium dynamics. Furthermore, we showed much faster equilibration is possible by including hydrodynamic interactions across a longer range. Hence, it is conceivable to use the more expensive Full HI approach to equilibrate the system, while using the cheaper NN HI and No HI schemes for equilibrium sampling.

Being based on first principles, and not model-dependent time-mapping, the presented method is in general applicable to other coarse-grained membrane models. If similar equilibrium kinetics were to be produced using explicit inclusion of solvent particles in a coarse-grained model, at least 10 solvent particles per membrane particle were necessary[73]. Even so, the kinetics can thus be investigated at a very small scale, because it is rather expensive to equilibrate even sub-micron-sized membrane patches[74]. Such an explicit solvent approach is impractical when considered on cellular scales. Meanwhile, we demonstrated the promise of our

approach in achieving large-scale simulations in biological systems, when various kinetics due to the membrane, cytoplasm, and extracellular environment are present. We observed, for example, a correlation between the haemoglobin concentration of red blood cells and the magnitude of membrane fluctuations. In previous experimental observation of such a correlation in diabetic patients, it has been argued that this effect is the result of glycation of haemoglobin and membrane proteins, altering both the cytoplasm and the membrane mechanics[65]. We have here reproduced such a correlation solely based on the different kinetics attributed to the cytoplasm. The correlation we observe can be explained by considering the fact that particles on the two membrane leaflets possess different mobilities, resulting in a different and orientation-dependent response to forces with either hydrodynamic or elastic origins. Thus, the free energy landscape, which describes the thermodynamics of the system of the red blood cell becomes, in general, dependent on the cytoplasmic viscosity (Fig. 6d). In other words, when applied to the complex system of the red blood cell, the various kinetic contributions are not trivially separable from the energy functional of our implicit-solvent model. While further investigations are necessary to establish a reliable link with experiments, we believe this outcome, from a detailed particle-based model, opens the door to potential cell-level investigations, with applications in biology and single-cell diagnosis.

## Methods
**Mesoscopic membrane model.** For all the simulations presented here, the bilayer is modelled as formed by particle dimers in a close-packed arrangement. The lattice parameter is in the range of 10 nm, while the two leaflets are resolved via the two particles forming the dimers[11]. The force field pertaining to bonded interactions is given by the following potentials[11],

$$U_s\left(r_{ij}\right) = D_e\left[1 - \exp\left(-\alpha\left(r_{ij} - r_{eq}\right)\right)\right]^2 \tag{7}$$

$$U_a\left(\theta_{i'ij}\right) = K_a\left(\theta_{i'ij} - \theta_{eq}\right)^2 \tag{8}$$

$$U_d(d_{ii'}) = K_d\left(d_{ii'} - d_{eq}\right)^2 \tag{9}$$

Particles belonging to each leaflet are connected to their nearest-neighbour counterparts via Morse-type bonds (Eq. (7)). Also, harmonic angle-bending potentials given by Eq. (8) act against the out-of-plane rotations of these bonds (the primed index designates the opposing particle in a dimer). Finally, particles in a dimer are connected via harmonic bonds of the form described by Eq. (9), which keeps the two leaflets together. Force-field parameters of $U_s$ and $U_a$ are obtained using a parameter-space optimization technique[11]. The optimization procedure minimizes the difference between the energy density resulting from the discrete interactions given by Eqs. (7) and (8) and the Helfrich energy density,

$$f_H = 2\kappa(H - H_0)^2 + \bar{\kappa}G \tag{10}$$

in which $\kappa$ and $\bar{\kappa}$ are the bending rigidity and Gaussian curvature modulus of the membrane, and $H$ and $G$, respectively, represent the mean and Gaussian curvatures. In addition, we have included the optimization of in-plane elasticity in the parametrization of the force field as well, via comparing the area compressibility modulus (Eq. (17)) with the experimental value. We chose the stiffness of the harmonic potential of Eq. (9), $K_d$, such that it prevents particles from flipping between the two leaflets, and results in stable bilayers for the long duration of simulations. The physical properties of the membrane used for parametrizing the force fields employed in equilibrium and non-equilibrium simulations of planar membranes are listed in Supplementary Table 1. The values of the force-field parameters, obtained or chosen for the simulations, are summarized in Supplementary Table 2.

Based on this model, the chemical composition of the membrane would reflect on the force field through the varying empirical properties used with this method. It is also possible to have heterogeneity in chemical composition, through the application of a non-uniform force field, locally representing the desired properties.

Finally, in order to model the in-plane fluidity of the membrane, we have implemented a procedure for updating the topology of bonded interactions in each simulation step via the so-called bond-flipping Monte Carlo moves[11,38]. One bond-flipping move comprises a proposal for switching a random in-plane bond to an intersecting one, and accepting or rejecting the switch based on the change in the potential energy using the Metropolis–Hastings algorithm[75].

**In-plane diffusion coefficient**. We use the Saffman–Delbrück model of the diffusion of cylindrical inclusions in fluid sheets to obtain the in-plane components, $D_i^{\parallel}$[41,42],

$$D_i^{\parallel} = \frac{kT}{4\pi\,\mu_{\mathrm{m}}d_{\mathrm{m}}}\left[\ln\left(\frac{\mu_{\mathrm{m}}d_{\mathrm{m}}}{\eta R_i}\right) - \gamma\right] \qquad (11)$$

where $R_i$ is the radius of a cylindrical inclusion in the membrane domain, $\mu_{\mathrm{m}}$ and $d_{\mathrm{m}}$ are the viscosity and thickness of the membrane domain, $\eta$ is the viscosity of the surrounding medium, and $\gamma \approx 0.577$ is the Euler–Mascheroni constant.

Note that a distinction should be made between the microscopic value of membrane viscosity, $\mu_{\mathrm{m}}$, used in Eq. (11), and the emerging macroscopic viscosity of the membrane, $\mu'_{\mathrm{m}}$, used, for example, in Eq. (16). The macroscopic viscosity is influenced by interactions and crowding effects, as well as specific in-plane dynamics. With our membrane model, bond-flipping Monte Carlo moves, that implement and control in-plane fluidity, significantly affect the resulting membrane viscosity[11,43]. We have thus calculated the macroscopic membrane viscosity, where needed, using the corresponding Green–Kubo relation[75,76],

$$\mu'_{\mathrm{m}} = \frac{A_{\mathrm{eq}}}{kT\,d_{\mathrm{m}}}\int_0^{\infty}\langle S_{xy}(\tau_0)S_{xy}(\tau_0+\tau)\rangle_{\tau_0}\,\mathrm{d}\tau \qquad (12)$$

where $A_{\mathrm{eq}}$ is the equilibrium projected area of the membrane, $S_{xy}$ is the in-plane shear stress defined as the shearing force per side length of the patch, and $\langle\cdots\rangle_{\tau_0}$ denotes ensemble averaging over starting times[76]. We have calculated the in-plane shear stresses during the simulation using the virial formula[77].

**Equilibrium undulations and dispersion relations**. We have defined the smooth function $h(\mathbf{r})$, with $\mathbf{r}=(x,y)$, as height function fitted to particle positions (Fig. 3a). Using fast Fourier transform, we find $h_{\mathbf{q}}(t)$, which is the amplitude of an undulatory mode with the wave vector $\mathbf{q}=\frac{2\pi}{L}(m,n)$, such that, $h(\mathbf{r})=\sum_{\mathbf{q}}h_{\mathbf{q}}\exp(i\,\mathbf{q}\cdot\mathbf{r})$. In thermal equilibrium, the ensemble average of the energy of these undulatory modes, for a membrane in a periodic box of side $L$, in the absence of any in-plane tensions, is given by the following relation[3],

$$\frac{1}{L^2}\langle h_{\mathbf{q}}h_{\mathbf{q}}^*\rangle = \frac{kT}{\kappa\,(qL)^4} \qquad (13)$$

this is a direct result of the Helfrich model (Eq. (10)) with the assumption of small-amplitude fluctuations[3,11]. This model thus predicts the amplitude of each undulatory mode to have the following distribution,

$$p\left(h_{\mathbf{q}}\right) \propto \exp\left(-\frac{\kappa L^2 q^4}{2\,kT}\left|h_{\mathbf{q}}\right|^2\right) \qquad (14)$$

In the continuum-based model of Seifert et al., in which the two membrane leaflets are resolved, and lipid density fluctuations, as well as inter-leaflet friction are present, the relaxation dynamics of undulatory modes has the following form[3,47,78],

$$\langle h_{\mathbf{q}}(t)h_{\mathbf{q}}^*(0)\rangle = A_1 e^{-\omega_1(q)t} + A_2 e^{-\omega_2(q)t} \qquad (15)$$

where the relaxation frequencies $\omega_1$ and $\omega_2$, are the eigenvalues of the time evolution operator, $-\boldsymbol{\Gamma}(q)\mathbf{E}(q)$, defined as,

$$\frac{\partial}{\partial t}\begin{pmatrix} h_{\mathbf{q}} \\ \rho_{\mathbf{q}} \end{pmatrix} = -\boldsymbol{\Gamma}(\mathbf{q})\mathbf{E}(\mathbf{q})\begin{pmatrix} h_{\mathbf{q}} \\ \rho_{\mathbf{q}} \end{pmatrix}$$

$$\boldsymbol{\Gamma}(\mathbf{q}) = \begin{pmatrix} \frac{1}{4\eta q} & 0 \\ 0 & \frac{q^2}{2\left(2b+2\eta q+\mu'_{\mathrm{m}}q^2\right)} \end{pmatrix}$$

$$\mathbf{E}(\mathbf{q}) = \begin{pmatrix} \tilde{\kappa}q^4 & -\frac{d_{\mathrm{m}}}{2}K_{\mathrm{area}}q^2 \\ -\frac{d_{\mathrm{m}}}{2}K_{\mathrm{area}}q^2 & K_{\mathrm{area}} \end{pmatrix} \qquad (16)$$

with $K_{\mathrm{area}}$ being the area compressibility modulus of the membrane, $\bar{\kappa}=\kappa+\frac{1}{4}d_{\mathrm{m}}^2K_{\mathrm{area}}$ the effective bending modulus, and $b$ the inter-leaflet friction coefficient. The quantity $\mu'_{\mathrm{m}}$ here denotes the macroscopic viscosity of the membrane obtained from Eq. (12).

To use Eq. (16) and obtain the theoretical eigenvalues $\bar{\omega}_1$ and $\bar{\omega}_2$, the viscosity of the solvent, as well as thickness, bending rigidity, and area compressibility modulus of the membrane, are all a priori given values used in the parametrization of the model (see "Mesoscopic membrane model" under Methods section). We have shown the bending rigidity to be very well reproduced by the membrane model (Fig. 3e–g and compare the power spectrum with the reference solid black line). The same agreement does not always hold true for the area compressibility modulus, and we have used actual values of this quantity for each simulation based on the fluctuations in the projected area of the membrane[79],

$$K_{\mathrm{area}} = \frac{kT\,A_{\mathrm{eq}}}{\langle A^2\rangle - \langle A\rangle^2} \qquad (17)$$

The only unknown quantity is the inter-leaflet friction coefficient, $b$. Experimental determination of $b$ is rather difficult, and is based on measuring the velocity

difference between the two leaflets in pulling tethers from vesicles[80,81]. The resulting values are in the range $10^8$–$10^9$ N s m$^{-3}$[80,82]. Yet, all-atom or coarse-grained simulations predict much smaller values in the $10^6$–$10^7$ N s m$^{-3}$ range[81,83,84]. It is to be expected that the inter-leaflet friction coefficient be highly sensitive to the resolution with which the lipids are modelled. Here, we have used the value of $b=10^7$ N s m$^{-3}$ to calculate $\bar{\omega}_{1,2}$ for all hydrodynamic models. In addition, we have added results corresponding to $b=10^6$ N s m$^{-3}$ to Fig. 3d, which yield a better fit for the slipping mode with the Full HI model. Both values of $b$ serve here as references, with their contribution being significant only to the slipping mode ($\bar{\omega}_1$).

**Active undulations**. To model active membranes, we have included representative particles in the model, that laterally diffuse in the membrane, while marking the positions where pump proteins are located and active forces are exerted. The UP and DOWN states of Eq. (6) correspond to forces being exerted in the two opposing normal directions, while the OFF state shows the absence of active forces (Fig. 5a). In reality, each pump protein can only exert a force in one direction, based on the way it is orientated in the membrane. The three-state kinetic model can be thought of as a symmetric extension of this behaviour, or a model for small clusters of proteins, in which the sum total of active forces can switch when different number of proteins are active. We assume the switching to occurs with equal rates $k_{\mathrm{on}}=k_{\mathrm{off}}=1/\tau_{\mathrm{p}}$.

We chose a rather large value of $F=3.7$ pN for the active forces to enhance the non-thermal effects, and assigned a timescale of $\tau_{\mathrm{p}}=500\,\mu$s to the switching reactions. Three active sites are considered in the model, and the membrane is coupled to an aqueous solvent similar to previous examples.

The mean-squared displacement of the membrane is defined as,

$$\overline{\langle h^2\rangle} = \frac{1}{L^2}\int_A\langle h^2(\mathbf{r})\rangle\mathrm{d}^2\mathbf{r} = \sum_{\mathbf{q}}\langle h_{\mathbf{q}}h_{\mathbf{q}}^*\rangle \qquad (18)$$

with $\overline{h^2}$ being the squared out-of-plane fluctuation amplitude, averaged over the projected area of the membrane, and $\langle\overline{h^2}\rangle$ is its ensemble average, calculated by sampling throughout the simulations. The summation in Fourier space is conducted for simulation trajectories after fitting the height function $h(\mathbf{r})$ to particle positions. To obtain theoretical expressions for the mean-squared displacement, we approximated the summation in Fourier space with an integral. Using Eq. (13) for a tension-free membrane at equilibrium, the mean-squared displacement is derived as,

$$\langle\overline{h^2}\rangle_{\mathrm{eq}} = \frac{kT\,L^2}{16\pi^3\kappa} \qquad (19)$$

For the non-equilibrium steady state, with the active forces following the three-state model of Eq. (6), we find the mean-squared displacement following the procedure laid out by Gov[52] and Lin et al.[54],

$$\langle\overline{h^2}\rangle_{\mathrm{active}} = \langle\overline{h^2}\rangle_{\mathrm{eq}} + p_{\mathrm{on}}\frac{F^2}{L^2}\frac{n_{\mathrm{p}}}{\sum_{\mathbf{q}}}\frac{\Lambda_{\mathbf{q}}^2}{\omega_{\mathbf{q}}}\left(\frac{1}{\omega_{\mathbf{q}}+g_{\mathbf{q}}+\tau_{\mathrm{p}}^{-1}}\right) \qquad (20)$$

where $\Lambda_{\mathbf{q}}=1/4\eta q$, $\omega_{\mathbf{q}}=\kappa q^3/4\eta$ and $g_{\mathbf{q}}=Dq^2$ with $D$ being the in-plane diffusion coefficient of active proteins. We obtained the value of $D$ from two-dimensional mean-squared displacements of active sites. Also, $F$ denotes the magnitude of the active force, $n_{\mathrm{p}}$ the surface density of active proteins, and $p_{\mathrm{on}}$ the probability of active proteins being on (in either of UP or DOWN states). Because of the analytical complexity of this expression, we carried out the integrals in Fourier space numerically.

**Simulation of planar membranes**. For simulations of equilibrium undulations and non-equilibrium relaxation of the membrane, planar membrane patches of 0.5 μm lateral size with in-plane periodic boundaries, and the lattice parameter of 10 nm were considered. Trajectories are obtained by updating the positions of particles according to Eq. (1). The diffusion tensor is updated in each integration step based on instantaneous normal vectors. Normal vectors are calculated for triangles formed between in-plane bonds and averaged for each particle based on its neighboring triangles. All the simulations are performed at $T=298$ K.

To obtain dispersion relations, we have performed simulations of membranes starting from a flat initial state, with respective trajectory lengths of 2.5 ms for the No HI and NN HI models, and 0.5 ms for the Full HI model. This choice is justified by looking at relaxation rates. In investigating kinetics of equilibrium membrane undulations, we were interested in equilibrium samples, and thus, we discarded an initial portion of each trajectory, allowing for complete equilibration. The length of the discarded portion was decided based on relaxation rates.

For the simulation of active membranes, we used membranes of 0.25 μm lateral size with periodic boundary conditions, with a lattice parameter of 5 nm, comparable with the size of a pump protein.

To sample from microstates describing a tension-free membrane, the Langevin piston barostat, coupled to the in-plane degrees of freedom, has been used[85]. Application of this barostat has the advantage of seamlessly fitting into the stochastic integrator already used for the in-plane degrees of freedom (Eq. (1)). Thus, the barostat parameters controlling the fluctuation timescale and dissipation

of the piston are chosen such that they represent a medium similar to the continuation of the membrane patch in the simulation box.

**Red blood cell**. Simulation setup for the red blood cell has been built using a 3D model obtained through refractive index tomography (courtesy of YongKeun Park[60,63,64] and Tomocube Inc., Republic of Korea). The model is captured at a mean resolution of ~100 nm, and consists of a triangular mesh with a wide size distribution. In order to apply the membrane model to this geometry, we have used two reference sets of force-field parameters, corresponding to lattice parameter values of 20 and 200 nm. We have employed an interpolation of the parameters to each bond, based on its neighbouring mesh size. Also, a correction based on the coordination number of particles is locally applied.

The properties of the red blood cell used in developing the model, apart from the elastic constants $\kappa$ and $\bar{\kappa}$, include the viscosity of the membrane, $\mu_m$, the cytoplasmic viscosity, $\eta_{cyt}$, pre-existing membrane surface tension, $\sigma_m$, the area compressibility modulus $K_{area}$ and the equilibrium volume, $V_{eq}$. The surface tension is applied by shifting the equilibrium distances of bonds compared with the lattice parameter dictated by the input mesh. As the membrane undergoes little internal reorganization on the scales of interest due to the presence of cytoskeleton, the bond-flipping Monte Carlo moves described in "Mesoscopic membrane model" under Methods section are not used. Volume preservation is enforced through a volumetric potential $U_v = K_{vol}\left(V - V_{eq}\right)^2$. Values of the input parameters used for parametrization, as well as the resulting force-field parameters are, respectively, given in Supplementary Tables 3 and 4.

We simulated the red blood cells for ~150 ms to obtain reliable statistics as well as large-scale kinetics. The baseline viscosity for the cytosol is taken to be $\eta_{cyt} = 4.5$ mPa s[71]. We considered the thickness profile of the red blood cell lying in the $xy$-plane to be given as $d_{RBC}(x, y)$, and obtained mean and mean-squared deviations of the thickness of the RBC, $\overline{d_{RBC}}$ and $\overline{\delta d_{RBC}^2}$, by sampling over the surface of the cell in each frame, to produce results in Fig. 6b–f. We have applied fast Fourier transform to the time series of the root mean-squared deviation of cell thickness, using the well-established Welch's method[86], to obtain power spectral density (PSD) of the thickness fluctuations shown in Fig. 6c and e.

Using the phase imaging data, we calculated thickness profiles (Fig. 6b) and power spectral densities of the thickness fluctuation of five healthy human red blood cells in an identical manner. Note that the experimental data on membrane fluctuations have been measured in vitro without ATP supply. Though no ATP-depleting treatments were carried out on the samples. The presence of active fluctuations is thus discouraged, but not completely excluded[60,65].

To calculate the change in the Helmholtz free energy of the red blood cell (Fig. 6d), using the expression $\Delta A = \Delta E - T\Delta S$, estimates of the internal energy, $E$, as well as the entropy, $S$ were needed. The internal energy was calculated based on the time averages of the sum of kinetic and potential energies sampled during the simulation. The potential energy is simply the sum of all bonded contributions, while the kinetic energy is estimated based the equipartition theorem. Note that we have implicitly assumed the irreversible evolution of the system to happen rather slowly, when using quasi-equilibrium assumptions for free energy calculations. Entropy estimation is done in Fourier space, using the time-dependent probability distribution corresponding to each vibration mode, and the expression $S = -k\sum_i p_i \log p_i$. Other contributions to the entropy from the side walls of the cell, or configurational changes below the resolution of the grid used for the fast Fourier transform, are thus ignored, and are assumed to stay constant for the duration of these calculations.

To relate the haemoglobin concentration to the viscosity of the cytoplasm, we have used a quadratic expression based on the experimental measurements of the rheology of haemoglobin solutions[87].

**Software**. Calculation of diffusion tensors are performed using codes developed in Python, with the help of SciPy computational package. Simulations based on the particle-based membrane model[11] are performed using an in-house specific-purpose software. The software is developed in C++, and multithreading parallelization is employed for enhanced performance.

The Python package Matplotlib is used for plotting the results. The software package Visual Molecular Dynamics (VMD) is used for some visualisations[88].

**Reporting summary**. Further information on research design is available in the Nature Research Reporting Summary linked to this Article.

## Data availability

The data that support the findings of this study are available from the corresponding authors upon reasonable request.

## Code availability

All the in-house developed software used in this study are either found at the public repository https://github.com/noegroup/membrane_kinetics, or available from the corresponding authors upon request.

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

## Acknowledgements

The authors wish to thank YongKeun Park and Geon Kim from department of physics, Korea Advanced Institute of Science and Technology (KAIST) for providing the three dimensional mesh and membrane fluctuation data of the human red blood cell. This research has been funded by Deutsche Forschungsgemeinschaft (DFG) through grants SFB 958/Project A04 "Spatiotemporal model of neuronal signalling and its regulation by presynaptic membrane scaffolds", SFB 1114/Project C03 "Multiscale modelling and simulation for spatiotemporal master equations", and European Research Commission, ERC CoG 772230 "ScaleCell".

## Author contributions

M.S. and F.N. designed research. M.S. conducted research and developed software. M.S. and F.N. wrote the paper.

## Competing interests

The authors declare no competing interests.
