## [Peer Review File · Nature Communications]

Peer Review File -

Reviewers' comments, round one:

Reviewer #2 (Remarks to the Author):

In this paper, a novel approach is presented to add a realistic description of kinetics to solvent free coarse-grained models of biomembranes. According to the authors "This paves the way for whole-cell simulations that still include nanometer/nanosecond spatiotemporal resolutions."

I very much doubt this will be the case. First of all, other groups have already combined solvent free coarse-grained models with hydrodynamic based approaches to improve the kinetic behavior of such models (e.g., Lyman and coworkers), and it is unclear why the current approach goes much beyond the existing ones. Second, even though the kinetic description gets more realistic, the underlying solvent free coarse-grain models remain very simplistic, and are unlikely to capture realistic dynamics to begin with.

My feeling is that the authors are overselling their method, which is a pity as the method itself might be useful in certain applications. I would therefore recommend to publish this work in a more specialized journal.

Below a point to point account of other specific points of criticism.

Subsection "Near-equilibrium kinetics of a planar membrane patch suspended in aqueous solvent":

- 1) I do not understand why the authors describe this section as "Near-equilibrium", while it is just the dynamics of the membrane fluctuations at equilibrium.
- 2) Fig3, e-g graphs, can the authors comment on the nature of the mismatch between the simulation results and prediction of the continuum model at high frequencies? This mismatch has been associated with, a) artifacts of binning and smoothing in real space before obtaining the Fourier spectrum. Brandt et al, Biophys. J, 100 2104-211 (2011), b) protrusion dominated regime, Lipowsky, et al. Biophys. Chem. 49:27-37 (1994), c) lipid tilt, May et al, PRE 76, 021913 (2007).
- 3) It seems that for fitting to $C(qL)^n$, there is a high-frequency cutoff in the fitting. Is this correct? and if it is, how is this cutoff determined? For "n" not equal to -4, which property "C" is described?
- 4) The authors attribute the presence of small deviations for long-wavelength modes in No HI and NN HI cases (high frequencies) to artefacts of hydrodynamic interaction cut-off. This is not really clear as the results show a slower relaxation for No HI and NN HI cases, therefore, long-wavelength modes relax slower and the deviation can be just due to low sampling as q^4 behavior is a feature of elastic energy (in absence of membrane tension), not the hydrodynamics.

Subsection "Far-from-equilibrium kinetics of the suspended membrane patch":

The system described in this section is just the relaxation of a membrane patch to equilibrium, and I cannot see any relation between the described system and far-from-equilibrium membranes, which usually includes continuous "energy input and dissipation" in which the system reaches a steady-state rather than an equilibrium state. See for example Ramaswamy et al, PRL 84 3494 (2000), Lin et al, J Chem. Phys 124, 074903 (2006), H. Turlier, T. Betz arXiv:1801.00176 and Lee et al, PNAS 114 (35) 9255-9260 (2017). I suggest that the authors extend their results section to include this type of system, to show the power of the method and increase the novelty of the article. Otherwise, all the results can be obtained from more detailed CG simulations.

Subsection "Fluctuation profile of a human red blood cell":

Flickering of red blood cells is indeed a prominent example of fluctuating biological membranes and its active nature has been precisely investigated, for example in Ref 88 (Turlier et al, Nature Physics 12, 513-519 (2016)). However, as the simulation does not contain any active force, I believe that the described behavior does not contain this important feature of red blood cells flickering (see H. Turlier, T. Betz arXiv:1801.00176). This issue needs clarification.

Reviewer #3 (Remarks to the Author):

This is a very elegant theoretical/computational work on how to efficiently model biomembranes. It is very nicely written and the examples that authors present are very impressive. I like this work very much, and I can see that this will have a strong impact on the field since the realistic simulations of membranes is still a big issue in the field.

I fully support the publication and I have only 1 comment:

1) It seems that the method applies for the membrane with all identical molecules, but real membranes are mixtures of different compounds. How this heterogeneity in chemical composition might affect the proposed method?

Reviewer #4 (Remarks to the Author):

The paper discusses a coarse-grained approach for replacing the solvent by continuum hydrodynamics to simulate lipid bilayer mechanics at large temporal and spatial scales. The aim is to capture correlation effects between lipids primarily during bending motions. A primary motivation is for the study of the fluctuation spectrum of red blood cells and the influence of different cytoplasmic viscosity arising from hemoglobin. The authors report results consistent with single-cell experimental observations. The presented work also provides a few studies of how hydrodynamic correlation effects influence lipid dynamics within membranes in general. This includes studies of the role of hydrodynamic effects in relaxation of undulations and results consistent with expected dispersion relations and the power spectral density (PSD) of membrane thickness fluctuations.

Overall, the paper is well-written and the figures provide clear illustrations of the methods and concepts. The work presents results clearly showing the influence of hydrodynamics on lipid membrane kinetics. The paper contributes potentially interesting new methods for simulating at the coarse-grained level lipid bilayer membranes over large temporal and spatial scales. However, there are a few items that should be addressed.

Specific Comments:

* The model is introduced in a way that is a bit unclear. In equation (1) the authors mix continuous time and discrete time notations. They also introduce the model in terms of a stochastic numerical discretization as opposed to clear underlying physical model in continuous time. Ideally, the authors should first introduce their model without reference the numerical methods / discretizations, then discuss how they will approximate the target physics.

* The discrete model as stated does not strictly obey detailed-balance. While this is common in numerical approximations, the authors should discuss how they determine the time-step and if they did any validation studies to make sure spontaneous drifts or other artifacts from the numerical approximations are kept small.

* In equation 2b, the Gaussian process $X(t)$ is not properly characterized in terms of moments. Again, because of the mixing of the discrete and continuous time notions, it should be a Kronecker delta-Function δ_{ij} not a Dirac delta-Function $\delta(t' - t)$.

* In the model the lipids appear to interact via hydrodynamics primarily through their normal component dynamics. While this might be a good approximation, it was unclear why the stress on the membrane surface from the fluid in the parallel directions were neglected. There are theories such as the Saffman-Delbruck results that show over large enough length-scales the external fluid stress plays a significant role. Perhaps the authors can comment on where this trade-off comes into play and why it can be neglected for the current system.

* In equation (4) only the single particle diffusion of the lipid in parallel direction is used from the Saffman-Delbruck theory for the planar case. However, one can also compute from the theory the two-point in-plane diffusion tensor, which for large viscosity is long-ranged. The authors should

discuss why they can neglect such interactions within the membrane.

* There is a subtlety in the model where the use of the diffusion tensors seems to double-count some of the contributions to membrane viscosity. The authors are resolving molecular level lipid-lipid interactions that already give rise to effectively a 2D liquid. It would be good for the authors to briefly discuss how they handle this issue and the parameters that contribute to the effective target viscosity for the membrane.

* The authors mention "we found a timestep of 0.5 ns to produce stable trajectories..." This should be clarified, since it seems the trajectories should be an accurate approximation of the underlying Langevin dynamics to ensure consistent with the target physics. It would seem the authors are treating the discretized approximation as the model. This might be alright from perspective of taking an experimentalist approach, but the authors should make this more explicit upfront in the paper and add some related discussions. For example, why is the 0.5 ns a good value beyond stability considerations. One could get stable looking simulation results that have poor properties with respect to statistical mechanics for example. The authors should discuss in more detail how the simulations and parameters are chosen to help ensure accurate and physically reliable results.

* In the PSD analysis of the red blood cell fluctuations, it would be helpful to add some more discussion of the underlying causes and implications of the shift in the spectrum. It is of interest that it appears to agree with trends seen in the experimental data set. There are a lot of potential contributing factors given the complexity of the model, but can the authors attribute some overall mechanism that explains the shift as the cytoplasmic viscosity is increased? It is mentioned that the spectrum can be used as a diagnostic. Can the authors attribute any interesting biophysical implications for red blood cells that would correlate with such observed shifts?

Reviewers' comments

Reviewer #2 (Remarks to the Author):

In this paper, a novel approach is presented to add a realistic description of kinetics to solvent free coarse-grained models of biomembranes. According to the authors "This paves the way for whole-cell simulations that still include nanometer/nanosecond spatiotemporal resolutions."

I very much doubt this will be the case. First of all, other groups have already combined solvent free coarse-grained models with hydrodynamic based approaches to improve the kinetic behavior of such models (e.g., Lyman and coworkers), and it is unclear why the current approach goes much beyond the existing ones. Second, even though the kinetic description gets more realistic, the underlying solvent free coarse-grain models remain very simplistic, and are unlikely to capture realistic dynamics to begin with.

My feeling is that the authors are overselling their method, which is a pity as the method itself might be useful in certain applications. I would therefore recommend to publish this work in a more specialized journal.

We did not intend to oversell our results and to suggest that our approach solves the cell-simulation problem, which is a vast enterprise with many challenges to be tackled. Rather, we suggest that we contribute an important and useful component in providing an explicit but coarse-grained membrane model that can reach greater length- and timescales as other methods that offer a similar level of resolution.

The highlight of our work, is to offer a computationally cheaper solution for including hydrodynamics, even in the local sense of anisotropic stochastic dynamics, and to show that combined with a highly coarse-grained model, this indeed makes it possible to perform tens- and even hundreds-millisecond-long simulations at the resolution offered by such models (nanometer range), while reproducing realistic kinetics. It is in this sense that we believe this work to be suitable for a larger and more general audience interested in such an approach for biophysical/biological research.

We have changed the introductory statements in the revised manuscript to tone down the statements that the reviewer perceived as overselling and clarified our contributions. On the other hand, to better demonstrate the promise of the proposed method, we added a new section on non-equilibrium steady state kinetics of active membranes (Sec. II D).

Below a point to point account of other specific points of criticism.

Subsection "Near-equilibrium kinetics of a planar membrane patch suspended in aqueous solvent":

1) I do not understand why the authors describe this section as "Near-equilibrium", while it is just the dynamics of the membrane fluctuations at equilibrium.

We agree with the reviewer about the unconventional naming of this and the next section, and have respectively renamed them to "Kinetics of a planar membrane at equilibrium" and "Non-equilibrium relaxation dynamics".

2) Fig3, e-g graphs, can the authors comment on the nature of the mismatch between the simulation results and prediction of the continuum model at high frequencies? This mismatch has been associated with, a) artifacts of binning and smoothing in real space before obtaining the Fourier spectrum. Brandt et al, Biophys. J, 100 2104-211 (2011), b) protrusion dominated regime, Lipowsky, et al. Biophys. Chem. 49:27-37 (1994), c) lipid tilt, May et al, PRE 76, 021913 (2007).

We thank the referee for pointing out this issue. The referee is right in that this deviation was indeed an artefact of the binning procedure used. We revised our code to minimize this artifact, and reproduced all the results using the revised code (Fig. 3). As can be verified, the new plots only deviate very slightly from the $1/q^4$ behavior for all frequencies.

3) It seems that for fitting to $C(qL)^n$, there is a high-frequency cutoff in the fitting. Is this correct? and if it is, how is this cutoff determined? For "n" not equal to -4, which property "C" is described?

With the revised approach to producing the power spectra (see previous comment), this issue is much less pronounced, considering the very good power-law fit to all the data points. Previously, we had indeed excluded high-frequency points, but this is no longer required with the new code.

4) The authors attribute the presence of small deviations for long-wavelength modes in No HI and NN HI cases (high frequencies) to artefacts of hydrodynamic interaction cut-off. This is not really clear as the results show a slower relaxation for No HI and NN HI cases, therefore, long-wavelength modes relax slower and the deviation can be just due to low sampling as q^4 behavior is a feature of elastic energy (in absence of membrane tension), not the hydrodynamics.

The referee is correct. In order to clarify this issue, we changed our simulation procedure, and allowed for a much longer equilibration time for the **No HI** and **NN HI** cases. The new results show negligible deviations at long-wavelength modes, and we have thus discarded our argument relating them to hydrodynamic cut-off.

Subsection "Far-from-equilibrium kinetics of the suspended membrane patch":

The system described in this section is just the relaxation of a membrane patch to equilibrium, and I cannot see any relation between the described system and far-from-equilibrium membranes, which usually includes continuous "energy input and dissipation" in which the system reaches a steady-state rather than an equilibrium state. See for example Ramaswamy et al, PRL 84 3494 (2000), Lin et al, J Chem. Phys 124, 074903 (2006), H. Turlier, T. Betz arXiv:1801.00176 and Lee et al, PNAS 114 (35) 9255-9260 (2017). I suggest that the authors extend their results section to include this type of system, to show the power of the method and increase the novelty of the article. Otherwise, all the results can be obtained from more detailed CG simulations.

We would like to thank the reviewer for pointing out this short-coming, and for the useful references. We investigated the applicability of the model in the non-equilibrium steady state kinetics of the so-called active membranes, as the reviewer had suggested. We renamed the section on relaxation dynamics to "Non-equilibrium relaxation dynamics", and added a new section on non-equilibrium steady state kinetics of membranes to include this additional investigation (Sec. II D "Non-equilibrium steady-state kinetics of active membranes"). Fig. 5 depicts the new results, which very well demonstrate the successful application of the method to the non-equilibrium steady state dynamics.

Subsection "Fluctuation profile of a human red blood cell":

Flickering of red blood cells is indeed a prominent example of fluctuating biological membranes and its active nature has been precisely investigated, for example in Ref 88 (Turlier et al, Nature Physics 12, 513–519 (2016)). However, as the simulation does not contain any active force, I believe that the described behavior does not contain this important feature of red blood cells flickering (see H. Turlier, T. Betz arXiv:1801.00176). This issue needs clarification.

We agree with the reviewer on the significance of these active fluctuations in human red blood cells. We have mentioned them in the first paragraph of the section on red blood cells. But we excluded them from our simulations, based on three justifications: (a) the active regime is effective on time-

scales beyond 100 ms. We have presented trajectories of at most 150 ms length, which while impressive for this particle-based model, fall drastically short of sampling such active contributions in the form of fluctuating forces. (b) Our main focus in this section is not to deeply delve into the complex world of red blood cell simulation, but to showcase how our kinetic approach can be applied to a complex membrane system, with different viscosities present, and to make useful quantitative predictions. (c) the experimental results that we used for the verification of the power-spectral density results have been produced *in-vitro* without ATP input (though without ATP-depletion). This discourages the presence of active fluctuations. We have amended Sec. II E to highlight these points, as follows:

...Our aim here is to show how different kinetics affects observables such as the magnitude of cell vibrations. Thus, while it is relatively straightforward to add active components to the model, similar to Sec. II D, we have refrained from doing so to reduce the complexity of the model. Also, as mentioned, these active contributions are important when timescales beyond 100 ms are considered, while the longest trajectories presented here, though of significant length for such a nanometer/nanosecond whole-cell simulation, can still be considered within this “passive” regime.

Also, the point (c) is explained in Methods section IV F,

Using the phase imaging data provided by Park *et al.*, we calculated thickness profiles (Fig. 6b) and power spectral densities of the thickness fluctuation of five healthy human red blood cells in a similar manner. Note that the experimental data on membrane fluctuations have been measured *in-vitro* without ATP supply. Though no ATP-depleting treatments were carried out on the samples. The presence of “active” fluctuations is thus discouraged, but not completely excluded [80, 85].

Reviewer #3 (Remarks to the Author):

This is a very elegant theoretical/computational work on how to efficiently model biomembranes. It is very nicely written and the examples that authors present are very impressive. I like this work very much, and I can see that this will have a strong impact on the field since the realistic simulations of membranes is still a big issue in the field.

I fully support the publication and I have only 1 comment:

1) It seems that the method applies for the membrane with all identical molecules, but real membranes are mixtures of different compounds. How this heterogeneity in chemical composition might affect the proposed method?

We would like to thank the reviewer for the kind remarks. The point raised by the reviewer is indeed very important, and to address it, we have added text to the Methods section, where the details of the membrane model are laid out. The short version is that the coarse-grained forcefield parameters (such as the ones given in Tab. II), strongly depend on the composition, and the heterogeneity in composition can be coupled to local force fields for convenient implementation.

Changes in Methods section IV A:

...Based on this model, the chemical composition of the membrane would reflect on the forcefield through the varying empirical properties used with this method. It is also possible to have heterogeneity in chemical composition, through the application of a non-uniform force field, locally representing the desired properties.

Reviewer #4 (Remarks to the Author):

The paper discusses a coarse-grained approach for replacing the solvent by continuum hydrodynamics to simulate lipid bilayer mechanics at large temporal and spatial scales. The aim is to capture correlation effects between lipids primarily during bending motions. A primary motivation is for the study of the fluctuation spectrum of red blood cells and the influence of different cytoplasmic viscosity arising from hemoglobin. The authors report results consistent with single-cell experimental observations. The presented work also provides a few studies of how hydrodynamic correlation effects influence lipid dynamics within membranes in general. This includes studies of the role of hydrodynamic effects in relaxation of undulations and results consistent with expected dispersion relations and the power spectral density (PSD) of membrane thickness fluctuations.

Overall, the paper is well-written and the figures provide clear illustrations of the methods and concepts. The work presents results clearly showing the influence of hydrodynamics on lipid membrane kinetics. The paper contributes potentially interesting new methods for simulating at the coarse-grained level lipid bilayer membranes over large temporal and spatial scales. However, there are a few items that should be addressed.

Specific Comments:

** The model is introduced in a way that is a bit unclear. In equation (1) the authors mix continuous time and discrete time notations. They also introduce the model in terms of a stochastic numerical discretization as opposed to clear underlying physical model in continuous time. Ideally, the authors should first introduce their model without reference the numerical methods / discretizations, then discuss how they will approximate the target physics.*

We agree with the reviewer that the notation was confusing. In the current manuscript, we restrict ourselves to the equations of the discretized version, which makes the paper better readable in our opinion. We decided against starting from the continuous equations in the current manuscript to put the emphasis on the application of the whole framework to large-scale simulations. A more detailed theoretical description, starting from the full Langevin description, is available in the following technical preprint: (M. Sadeghi and F. Noe (2019) *arXiv* 1909.02722, <https://arxiv.org/abs/1909.02722>).

** The discrete model as stated does not strictly obey detailed-balance. While this is common in numerical approximations, the authors should discuss how they determine the time-step and if they did any validation studies to make sure spontaneous drifts or other artifacts from the numerical approximations are kept small.*

The reviewer has raised a valid point regarding the artefacts of approximate detailed balance observation. In order to verify the goodness of our approximate, large-timestep method, we validated the sampling of the fastest dynamical mode in the system (smallest wavelength mode in equilibrium fluctuations). As can be seen in the inset plots of Figs. 3(e)-3(g), our approach has resulted in correct sampling of this mode, and the Gaussian profile predicted by the Helfrich functional is very well reproduced.

** In equation 2b, the Gaussian process $X(t)$ is not properly characterized in terms of moments. Again, because of the mixing of the discrete and continuous time notions, it should be a Kronecker delta-Function δ_{ij} not a Dirac delta-Function $\delta(t' - t)$.*

Thank you for pointing out this issue, we have corrected the given moments of the Gaussian process in the manuscript.

** In the model the lipids appear to interact via hydrodynamics primarily through their normal component dynamics. While this might be a good approximation, it was unclear why the stress on the membrane surface from the fluid in the parallel directions were neglected. There are theories such as the Saffman-Delbruck results that show over large enough length-scales the external fluid stress plays a significant role. Perhaps the authors can comment on where this trade-off comes into play and why it can be neglected for the current system.*

We believe the shearing contributions to be important where they are comparable to in-plane viscous forces existing in the coarse-grained model. This is clearly scale-dependent, i.e. depends on the resolution of the membrane model, and we believe that in case of our membrane model, where the diffusion of membrane bound species is dictated by prescribed in-plane mobility of large beads, as well as bond-flipping Monte Carlo moves at 5-10nm scale, such solvent-mediated contributions are negligible. To address the reviewer's point, and to clarify our argument, we have modified the following text in Sec. II A,

...The main contribution to solvent-mediated hydrodynamic forces acts along the membrane normal. Shearing interactions via the solvent can indeed affect in-plane diffusion [56, 57], but are generally dominated by much larger in-plane viscous forces, especially in highly coarse-grained models. Also, they can be neglected, when large-scale out-of-plane dynamics are considered. Finally, if there are other mechanisms controlling the in-plane diffusion, such as the bond-flipping Monte Carlo moves [22, 65], the contribution from shearing interactions becomes redundant.

** In equation (4) only the single particle diffusion of the lipid in parallel direction is used from the Saffman-Delbruck theory for the planar case. However, one can also compute from the theory the two-point in-plane diffusion tensor, which for large viscosity is long-ranged. The authors should discuss why they can neglect such interactions within the membrane.*

We have explained the reason why we did not include in-plane hydrodynamics in Sec. II A,

...While in-plane hydrodynamics of bilayer membranes can also be studied rigorously [58, 59], a highly coarse-grained membrane model would benefit little from it. Also, for in-plane diffusion in a membrane crowded with proteins, there is evidence pointing to the hydrodynamics being effectively reduced to a collision-based dynamics, resulting in a Stokes-Einstein-like diffusion [60].

** There is a subtlety in the model where the use of the diffusion tensors seems to double-count some of the contributions to membrane viscosity. The authors are resolving molecular level lipid-lipid interactions that already give rise to effectively a 2D liquid. It would be good for the authors to briefly discuss how they handle this issue and the parameters that contribute to the effective target viscosity for the membrane.*

In our specific membrane model, which has been used to produce the results here, the in-plane fluidity is a direct result of bond-flipping Monte Carlo moves and not molecular level lipid-lipid interactions. But in general, the in-plane component of the diffusion tensor, as well as the frequency of bond-flipping moves, affect the macroscopic viscosity of the membrane. To clarify this distinction, we have used two notations for the "microscopic" vs. "macroscopic" viscosities of the membrane, and have shown how the macroscopic viscosity can be calculated used Green-Kubo relation,

Methods section IV B,

...Note that a distinction should be made between the microscopic value of membrane viscosity, μ_m , used in Eq. (8), and the emerging macroscopic viscosity of the membrane,

μ'_m , used, for example, in Eq. (13). The macroscopic viscosity is influenced by interactions and crowding effects, as well as model-specific in-plane dynamics. With our membrane model, bond-flipping Monte Carlo moves, that implement and control in-plane fluidity, significantly affect the resulting membrane viscosity [22, 63]. We have thus calculated the macroscopic membrane viscosity, where needed, using the corresponding Green-Kubo relation [103, 104],...

we have also revised the code used for calculation of the Seifert model dispersion relation to make sure that the actual macroscopic membrane viscosity is used to produce Figs. 5(b)-5(d).

** The authors mention “we found a timestep of 0.5 ns to produce stable trajectories...” This should be clarified, since it seems the trajectories should be an accurate approximation of the underlying Langevin dynamics to ensure consistent with the target physics. It would seem the authors are treating the discretized approximation as the model. This might be alright from perspective of taking an experimentalist approach, but the authors should make this more explicit upfront in the paper and add some related discussions. For example, why is the 0.5 ns a good value beyond stability considerations. One could get stable looking simulation results that have poor properties with respect to statistical mechanics for example. The authors should discuss in more detail how the simulations and parameters are chosen to help ensure accurate and physically reliable results.*

As the reviewer suggested, we looked at the quality of equilibrium sampling for the rather large timestep used here. To do so, we considered the fastest dynamical mode of the system (out-of-plane motion of smallest wavelength fluctuations) and showed the resulting distribution to exactly match the predicted equilibrium distribution (inset plots of Figs. 3(e)-3(g)).

** In the PSD analysis of the red blood cell fluctuations, it would be helpful to add some more discussion of the underlying causes and implications of the shift in the spectrum. It is of interest that it appears to agree with trends seen in the experimental data set. There are a lot of potential contributing factors given the complexity of the model, but can the authors attribute some overall mechanism that explains the shift as the cytoplasmic viscosity is increased? It is mentioned that the spectrum can be used as a diagnostic. Can the authors attribute any interesting biophysical implications for red blood cells that would correlate with such observed shifts?*

We agree that this point needed more discussion. Thus, we have added the following to the text in Sec. III,

...The correlation we observe can be explained by considering the fact that particles on the two membrane leaflets possess different mobilities, resulting in a different and orientation-dependent response to forces with either hydrodynamic or elastic origins. Thus, the free energy landscape, which describes the thermodynamics of the system of the red blood cell becomes, in general, dependent on the cytoplasmic viscosity (Fig. 6d). In other words, when applied to the complex system of the red blood cell, the various kinetic contributions are not trivially separable from the energy functional of our implicit-solvent model.

As for the potential applications, we mentioned a direct diagnosis relevant to this effect (Diabetes mellitus), in Sec. II E,

...The seemingly linear correlation between the membrane fluctuation and Hb concentration (Fig. 6f) has been previously suggested based on quantitative phase imaging techniques [80, 85]. Specifically, Lee et al. have shown that in patients with diabetes mellitus, the change in the concentration of hemoglobin in red blood cells has a statistically significant effect on membrane fluctuations, with a correlation very similar to Fig. 6f [85].

REVIEWERS' COMMENTS round two:

Reviewer #2 (Remarks to the Author):

The authors have improved their manuscript by including novel results and modifying the text. I am happy to recommend it for publication in Nature Comm. after the following two comments are being addressed.

1) The authors state "When active forces are present in the simulation, the height distribution deviates visibly for the large wavelength modes (Fig. 5b). This deviation from the Gaussian distribution is the hallmark of active processes [77], and is the result of active forces constantly driving the system out of equilibrium". Can the authors quantitatively show that the height distribution is deviating from a Gaussian distribution? Figure 5b only shows deviation from an equilibrium distribution.

2) How can the authors distinguish their results (for the active membrane) from the passive softening effects (curvature instability)? For example: for active systems, q^{-4} behavior is violated while for passive softening q^{-4} behavior is preserved. See work by Pezeshkian et al, Soft Matter 15, 9974 2019.

Reviewer #4 (Remarks to the Author):

The revisions address most of the comments.

Response to Reviewer Comments:

Reviewer #2 (Remarks to the Author):

The authors have improved their manuscript by including novel results and modifying the text. I am happy to recommend it for publication in Nature Comm. after the following two comments are being addressed.

1) The authors state "When active forces are present in the simulation, the height distribution deviates visibly for the large wavelength modes (Fig. 5b). This deviation from the Gaussian distribution is the hallmark of active processes [77], and is the result of active forces constantly driving the system out of equilibrium". Can the authors quantitatively show that the height distribution is deviating from a Gaussian distribution? Figure 5b only shows deviation from an equilibrium distribution.

Author response:

We tested the non-Gaussianity of distributions in the active cases via the D'Agostino-Pearson normality test and have added the corresponding K-squared values for all the distributions shown in Fig. 5b. The K-squared test, based on skewness and kurtosis of the samples, clearly demonstrates the deviation from the Gaussian distribution when active forces are present.

2) How can the authors distinguish their results (for the active membrane) from the passive softening effects (curvature instability)? For example: for active systems, q^{-4} behavior is violated while for passive softening q^{-4} behavior is preserved. See work by Pezeshkian et al, Soft Matter 15, 9974 2019.

The active agents in our simulations only exert forces normal to the membrane surface and have no curvature-coupling. Thus, we believe that the mentioned softening effects could not be present. Any deviation from the equilibrium distribution is solely due to the response to the active forces. The quantitative comparison of mean squared displacement between theory and simulation, presented in Fig. 5c, is based on the non-equilibrium theory developed by Gov (2004) and Lin et al. (2007) (briefly reproduced in the Methods section III D). It directly correlates membrane

displacement with the magnitude of the force, the kinetics of the switching behaviour, and the inplane density and diffusion coefficient of active agents. The accurate match between the theory and the simulation results, as seen in Fig. 5c, points to the fact that the non-equilibrium effects are indeed modelled as expected.

Reviewer #4 (Remarks to the Author):

The revisions address most of the comments.

We thank the reviewer again for the very useful comments on the first round of review.